# Tumoral C2 Regulates the Tumor Microenvironment by Increasing the Ratio of M1/M2 Macrophages and Tertiary Lymphoid Structures to Improve Prognosis in Melanoma

**DOI:** 10.3390/cancers16050908

**Published:** 2024-02-23

**Authors:** Gengpu Zhang, Shengnan Li, Wanyi Xiao, Chao Zhang, Ting Li, Zhichao Liao, Haotian Liu, Ruwei Xing, Wei Yao, Jilong Yang

**Affiliations:** 1Department of Bone and Soft Tissue Tumor, Tianjin Medical University Cancer Institute and Hospital, National Clinical Research Center for Cancer, Key Laboratory of Cancer Prevention and Therapy, Tianjin’s Clinical Research Center for Cancer, Tianjin 300060, China; zgp840123111@tmu.edu.cn (G.Z.);; 2Department of Oncology, The Fifth Affiliated Hospital, Sun Yat-Sen University, Zhuhai 519000, China; lsn@tmu.edu.cn

**Keywords:** melanoma, complement C2, macrophage, tertiary lymphoid structures, tumor microenvironment

## Abstract

**Simple Summary:**

Melanoma is an immunogenic tumor, so its outcomes and the efficacy of immunotherapy must be linked to the immune component. In this study, we aimed to find indicators that would allow us to monitor both the prognosis of patients and the efficacy of immunotherapy. We found that complement C2 tended to be upregulated in patients after effective anti-PD-1 therapy, and the high expression group achieved longer survival. Tumor tissues with C2 expression were infiltrated with more M1-type macrophages and tertiary lymphoid structures, and this favorable immune status may be responsible for the favorable outcome. Our findings provide valuable insights into the functional role of C2 in melanoma and its potential implications for clinical therapy.

**Abstract:**

Immunotherapy is an essential therapy for individuals with advanced melanoma. However, not all patients respond to such treatment due to individual differences. We conducted a multidimensional analysis using transcriptome data from our center, as well as publicly available databases. We found that effective nivolumab treatment led to an upregulation of C2 levels, and higher levels following treatment are indicative of a good outcome. Through bioinformatics analyses and immunofluorescence, we identified a correlation between C2 and M1 macrophages. To further investigate the role of C2 in melanoma, we constructed subcutaneous tumorigenic models in C57BL/6 mice. The tumors in the C2 overexpression group exhibited significantly smaller sizes. Flow cytometric analysis of the mouse tumors demonstrated enhanced recruitment of macrophages, particularly of the M1 subtype, in the overexpression group. Moreover, single-cell RNA sequencing analysis revealed that C2-positive tumor cells exhibited enhanced communication with immune cells. We co-cultured tumor cell supernatants with macrophages in vitro and observed the induction of M1 subtype polarization. In addition, we discovered a close correlation between C2 and tertiary lymphoid structures. C2 has been demonstrated to exert a protective effect, mediated by its ability to modulate the tumor microenvironment. C2 serves as a prognostic marker for melanoma and can be employed to monitor the efficacy of immunotherapy.

## 1. Introduction

The malignant alteration of pigment-producing melanocytes in the basal layer of the epidermis leads to the development of melanoma. More than 75% of skin cancer deaths are caused by melanoma, even though it only causes 5% of skin cancer cases overall [1]. In contrast to cold tumors like pancreatic cancer, melanoma has significant immunogenic features, and immunotherapy, in addition to standard therapies like surgery, chemotherapy, and targeted medications, has proven helpful in increasing the survival of melanoma patients [2]. Nevertheless, a portion of melanoma patients continue to have little improvement from immunotherapy.

Melanoma tumor cells interact with their environment to create a diverse tumor microenvironment [3]. An improved prognosis is linked to elevated levels of stromal cells, particularly tumor-infiltrating lymphocytes [4]. In addition, the interaction between lymphocytes and tumor cells within the tumor microenvironment (TME) determines the effectiveness of the immunotherapy [5,6]. It is worth noting that besides cellular immunity, humoral immunity also plays a crucial role. The relationship between complements and cancer has become an emerging field, and it has been demonstrated in investigations into several cancer species that complements can influence the course of tumor development [7]. However, little has been reported about complements and melanoma, and the research available is either conducted in vitro or based on animal models. A significant amount of research concentrates on the crucial nodes C1q, C3, and C5, as they are involved in the cancer promotion process [8,9,10]. Among the molecules upstream of the classical complement activation pathway, the involvement of C2 in tumorigenesis remains unknown.

We discovered for the first time that C2 transcriptome levels continuously increase during effective anti-PD-1 immunotherapy. Furthermore, higher C2 levels after immunotherapy could predict better prognosis for patients. Thus, we further investigate the C2 and tumor microenvironment in our current study. Not only do samples with high levels of C2 exhibit superior immune states in human melanoma, but the development of tertiary lymphoid structures (TLSs) is also correlated with C2 levels. In clinical practice, there is a need for a simple indicator to assess the immune status of melanoma that can predict prognosis and dynamically monitor the efficacy of the administered immunotherapy. Fulfilling this purpose, C2 is expected to be integrated with the current treatment guidelines to formulate novel strategies.

## 2. Materials and Methods

### 2.1. Data Sources and Preprocessing

The Cancer Genome Atlas’s (TCGA) Skin Cutaneous Melanoma (SKCM) study provided bulk RNA-seq and clinical information [11]. Exclusion criteria were as follows: (1) patients whose tissue type was unclear according to the tissue number, such as number 8744/3; (2) patients who had preoperative neoadjuvant therapy and radiotherapy; (3) patients who had preoperative systemic therapy. There were 430 tumor specimens available. Before the correlation analysis and Cox regression analysis, we ran a second round of case screening on patients who had more than three missing clinicopathological items. In addition, GSE65904, which held prognostic data belonging to 214 individuals with melanoma, was employed [12]. The single-cell expression profile of melanoma, which is capable of demonstrating the global immune landscape, is available in GSE174401 [13]. The transcriptional data before and after treatment with nivolumab were derived from the CA209-038 study [14]. In our single-center trial, frozen tissue extracted from 57 melanoma individuals was utilized for RNA sequencing. The resulting raw bulk RNA sequencing data had been previously published and recorded under accession code GSE215121 [15].

### 2.2. Cell Culture and Transfection

The human melanoma cell line A375 (Beijing Cellular Research Institute, Beijing, China) was cultured in DMEM (Invitrogen, Carlsbad, CA, USA) with 10% fetal bovine serum added. Murine melanoma cell line B16 and human macrophage cell line THP-1 were kindly donated by Prof. Jun Li’s laboratory and cultured in RMPI-1640 medium (Invitrogen) and maintained at 37 °C with 5% CO_2_. THP-1 cells are initially stored in suspension and require the addition of Phorbol 12-myristate 13-acetate (PMA) for wall attachment. Human complement C2 cDNA (NM_000063.5) and mouse c2 cDNA (NM_013484.2) were separately cloned into a pCDH plasmid expression vector. pCDH vector was used as a control. Standard protocols were followed for performing lentiviral infections.

### 2.3. RNA-Seq Data Analysis

Setting logarithmic fold changes larger than 1.5 and *p*-values < 0.05, the DESeq2 and edgeR tools were used to examine the differentially expressed genes (DEGs) within the C2^high^ and C2^low^ groups, respectively. We evaluated pathways connected to DEGs via the gene ontology (GO) and Kyoto Encyclopedia of Genes and Genomes (KEGG) databases [16].

### 2.4. Single-Cell RNA-Seq Filtrating and Processing

We utilized the Seurat algorithm (v4.0.0) [17] for single-cell quality control, data normalization, downscaling, and unsupervised clustering to construct a global atlas. In particular, we investigated single-cell transcriptome information collected from samples belonging to eight patients having cutaneous melanoma and fourteen patients with brain metastases. Cells were eliminated if less than 200 or more than 5000 genes were found. Moreover, low-quality cells were removed if they had erythroid cells larger than 3%, mitochondrial genes greater than 15%, or fewer than 1000 unique molecular identifiers (UMIs) per cell.

### 2.5. Tumor Microenvironment and Immune Infiltrate Analysis

ESTIMATE analysis was employed to determine the TME component by calculating the immune score for each sample [18]. The proportion of immune subsets invading the tumor sample was determined with CIBERSORT, relying on a built-in signature matrix with defined immune cells [19]. The TIMER 2.0 online tool was used to determine whether RNA levels and immune cell infiltration ratings were correlated. Twenty-four immune cell types were measured utilizing single-sample gene set enrichment analysis (ssGSEA) based on GSVA [20]. According to the median C2 expression level, all samples were split into two groups and then contrasted.

### 2.6. Cell–Cell Interaction Analysis

Using CellChat, it is possible to statistically predict and evaluate intercellular communication networks via scRNA-seq to determine how macrophages, T cells, and tumor cells interact with each other within tissues [21]. The interactions between the receptors and ligands of two cell clusters were identified according to the unique expression of receptors by one cell subpopulation and ligands by another.

### 2.7. Pharmacotherapy Prediction Response

The OncoPredict program predicts the IC50 value of a sample based on the cell line’s sensitivity to the drug and gene expression [22]. TIDE employs a collection of gene expression indicators to assess two types of tumor immune escape mechanisms: some immunosuppressive factors preventing T cell infiltration and high levels of functionally inactivated T cell infiltration. Patients with a high TIDE score had poor immune checkpoint blockade effectiveness and short post-ICB survival [23].

### 2.8. Immunohistochemical Staining, Immunofluorescence, and Immunocytochemistry

Paraffin-embedded tissue blocks from individuals with surgically resected melanoma between 2010 and 2013 in Tianjin Medical University Cancer Hospital were used in this study. Primary antibodies were incubated on blocked sections for a whole night at 4 °C before a 30 min incubation with the secondary antibodies at 37 °C. The primary antibodies were used in experiments including anti-CD8A (ZSGB-Bio, Beijing, China, ZA-0508), anti-CD4 (ZSGB-Bio, ZA-0519), anti-CD20 (Proteintech, Rockford, IL, USA, 60271-1-Ig), and anti-C2 (Santa Cruz Biotechnology, sc-373809). Positive cell presence was scored from 0 to 4, with 0 meaning a lack of positive cells, and 4 indicating over 75% positive cells. The staining intensity was categorized using the following criteria: 1 denotes no staining, 2 is mild staining, 3 is moderate staining, and 4 is high staining. Staining intensity score × percentage of positive tumor cells was the formula used to determine the staining index (SI). The high and low expression groups were then divided by eight.

Multiplex immunofluorescence staining was performed utilizing the Opal^TM^ multi-Color Manual IHC Kit (PerkinElmer, Waltham, MA, USA) according to the instruction manual with the following primary antibodies: C2 (Santa Cruz Biotechnology, Santa Cruz, CA, USA, sc-373809), CD86 (Abcam, Cambridge, UK, ab239075), CD206 (CST, E6T5J), Ki-67 (Abcam, ab16667), and CD163 (Abcam, ab182422).

Cells on chamber slides were treated with 4% paraformaldehyde for twenty minutes and incubated with primary antibodies against CD86 (Abcam, ab239075) and CD163 (ZSGB-Bio, TA506380). After counterstaining with DAPI, panoramic imaging equipment was used to scan the image.

### 2.9. Western Blot

The cell lysates were prepared with RIPA buffer. Antibodies specific to the following molecules were used: C2 (Abcam, ab209900) and GAPDH (Proteintech, 60004-1-Ig). Equal quantities of total protein were present in the cell lysates that were electrophoresed in 10% sodium dodecyl sulfate–polyacrylamide gels before being transferred to nitrocellulose membranes. The membranes were blocked for two hours with 5% milk and incubated with the primary antibody overnight. After that, the membrane was soaked in TBST three times and allowed to incubate for one hour with the secondary antibody. Protein bands were detected through the use of enhanced chemiluminescence.

### 2.10. Mouse Models

C57BL/6 immunocompetent mice aged 6 weeks were purchased from Vital River Laboratory Animal Center (Beijing, China). The laboratory mice were kept in pathogen-free conditions. The animal experimental procedures were approved by the Ethics Committee of Tianjin Medical University Cancer Institute and Hospital. Every experimental group had a randomization. Subcutaneous injections of 2 × 10^6^ B16 tumor cells were given to each animal. Calipers were used to track the development of the tumor, and the volume was computed using the following formula: volume = 1/2 L1 × (L2)2.

### 2.11. RT-PCR

TRIzol was used to extract total RNA, and TransGen Biotech’s reverse transcription reagent converted the RNA to cDNA. SYBR Green master mix was utilized for qPCR, and β-actin was employed as a control. The primers are shown in Appendix A.

### 2.12. Cell Counting Kit-8 (CCK8) Assay

Equal-density cells were seeded into 96-well plates with three replicate wells for each cell. Tumor cells were incubated with CCK8 reagent for 2 h in the absence of light, and then, the absorbance at 450 nm was measured each day using a microplate reader. A growth curve was plotted based on absorbance.

### 2.13. Flow Cytometry

The milled cells were resuspended in 1% BSA and treated with anti-mouse CD45 (BioLegend, San Diego, CA, USA, 147713), CD11b (BD Biosciences, Franklin Lakes, NJ, USA, 612977), F4/80 (BioLegend, 123141), CD80 (BD Biosciences, 562611), and CD163 (Invitrogen, 63-1631-82) and CD206 (BD Biosciences, 568808) for analysis of surface markers. Human anti-CD86 (BD Biosciences, 747525) is used in the THP-1 cell assay. Every antibody was used at the concentration suggested by the package insert. Multi-color FACS analysis was performed on a Beckman flow cytometer. All flow cytometry results were processed through FlowJo.

### 2.14. Preparation of Conditioned Medium (CM) and Macrophage Induction

A375 cancer cells (20 × 10^4^ cells/well) were inoculated in 6-well plates for 24 h. The supernatant was collected for macrophage processing, and the supernatant was filtered through a filter with a 0.2um pore size before use. To induce THP-1 macrophage polarization, the CM was replaced once after 36 h.

### 2.15. Statistical Analysis

The nonparametric unpaired Wilcoxon test was used to compare changes. The Chi-square test and Fisher’s test were applied to the correlation analysis of dichotomous variables. Spearman’s correlation analysis was employed in the continuous variable correlation analysis. The survminer R package (v0.4.9) was used to perform survival analysis. A statistically significant difference was defined as *p* < 0.05.

## 3. Results

### 3.1. The Clinical Value of Complement C2 in Melanoma

Nivolumab, a classic PD-1 inhibitor, has been applied in clinical practice, and many melanoma patients achieved stable disease (SD), partial response (PR), or even complete response (CR) after effective nivolumab treatment in the CA209-038 study. The expression of C2 showed a trend of upregulation during nivolumab treatment, whereas C2 exhibited no significant changes or even partial downregulation in the immunotherapy failure group (Figure 1A). There was no significant difference in pre-treatment baseline between progressive disease (PD) and non-PD groups (Appendix A). Furthermore, patients with high C2 expression after effective immune therapy exhibited longer survival times (Figure 1B). We then further explored the clinical significance of C2. The relationship between clinicopathological items and C2 was analyzed in the TCGA dataset. Our findings confirm a negative correlation between C2 and ulceration formation, which is a carcinogenic process (Table 1). C2 level could therefore be a suitable prognosticator, as it was shown to independently impact overall survival (OS) in multi-Cox analysis (HR = 0.53, 95% CI: 0.28-0.99, *p* = 0.047, Table 2). In three independent datasets, including our center’s internal sequencing data, the findings revealed that patients with high C2 levels showed a more favorable survival prognosis, further confirming its clinical significance (Figure 1C,D and Appendix A). Consistent with transcription level, patients with higher C2 protein expression showed a more favorable outcome among 73 cases (Figure 1E,F). In pan-cancer survival analyses, high levels of C2 in not only melanoma but also lung adenocarcinoma (LUAD) and liver hepatocellular carcinoma (LIHC), which are cancers with good immunogenicity, also indicated a better prognosis (Appendix A). In summary, complement C2 plays a crucial protective role in the disease course and the effectiveness of immunotherapy.

### 3.2. Complement C2 Implies a More Advantageous Immune Status

We identified the DEGs between the two groups using the C2 median as the cutoff, covering 976 upregulated genes (Figure 2A and Appendix A). The functional enrichment results including GO annotation and KEGG analysis suggest that C2 is inextricably linked to the tumor immune environment (Figure 2B and Appendix A). “Antigen processing and presentation”, “Phagosome”, “Toll-like receptor signaling”, and “JAK-STAT signaling” suggest that macrophages were oriented towards the M1 type in context. In addition, “B cell receptor signaling” was enriched. The immune scores calculated by ESTIMATE for the high C2 samples were significantly higher, indicating a greater infiltration of immune components (Figure 2C). Immune component analysis revealed that macrophages and T cells were the major components of melanoma. Melanoma tissue with higher C2 levels had a significantly higher proportion of M1 macrophages and fewer M2 macrophages, as well as increased levels of CD8+ T cells, activated CD4+ T cells, and plasma cells (Figure 2D). Subsequent immunofluorescence analysis in twenty-one individuals demonstrated that CD86+ macrophages were scattered in the high C2 expression regions, while CD163+ macrophages were more likely to be distributed in the low C2 expression regions (Figure 2E, Table 3). In addition, there is a negative correlation between the expression level of C2 and CD206+ macrophages in tumor tissue (Appendix A). One of the reasons why C2 affects the prognosis and immunotherapy efficacy of melanoma is that it can impact the TME of melanoma.

### 3.3. C2 Inhibits Melanoma Growth and Regulates M1/M2 Macrophage Ratio In Vivo

We prepared mouse melanoma cells (B16) stably expressing C2 (Figure 3A and Appendix A) and established subcutaneous tumor models in C57BL/6 mice to explore the effect of tumor-derived C2 on macrophages in the TME (Figure 3B). We found that the overexpression of C2 did not affect melanoma proliferation in vitro (Appendix A). However, C2-overexpressing tumors exhibited significant growth retardation in vivo compared to the vector mice (Figure 3C–E and Appendix A). We then analyzed macrophage subpopulations in each group of mouse tumors using flow cytometry. We progressively detected the surface marker CD80 associated with M1 macrophages and CD163 associated with M2 macrophages to clarify the proportions of tumor macrophage subtypes (Figure 3F). Macrophages labeled by F4/80+CD11b+CD45+ were significantly more abundant in C2-overexpressing tumors (Figure 3G,H). At the same time, macrophages in C2-overexpressing tumors favored the M1 phenotype with increased M1-type macrophages (CD80+CD163− and decreased M2-type macrophages (CD80−CD163+) compared to controls (Figure 3I–K). Furthermore, we confirmed that C2-overexpressing tumors result in decreased infiltration of CD206+ macrophages (Appendix A). All this suggests that C2 does not directly determine melanoma proliferation but that it inhibits cell growth in vivo, characterized by an increased polarization of M1 macrophages and decreased polarization of M2 macrophages. The reason C2 can inhibit melanoma growth in vivo is likely related to C2′s ability to recruit more macrophages and regulate the M1/M2 subtype ratio.

### 3.4. Tumoral C2 Promotes Macrophages Differentiation into the M1 Subtype In Vitro

Single-cell sequencing technology is an effective approach to understanding the TME. The spatial distribution of tumor and stromal cells is visualized based on cell-specific markers (Figure 4A and Appendix A). The interactions of ligands and receptors and the activation of certain cell-signaling pathways are the foundations of cell communication. C2-positive tumor cells have more prominent communication with immune cells than C2-negative cells do (Figure 4B). Interaction between CD4+/CD8+ T cells and C2-positive tumors is associated with IFNG-IFNGR and GZMA-F2R, enhancing T cell killing. In addition, cancer-promoting ligands and receptors (SPP1-CD44) are the key mediators of interactions between C2-negative tumor cells and immune cells, which supports M2 polarization and promotes tumor progression (Figure 4C). We next evaluated whether C2 can induce macrophage polarization in cultured cells. Human THP-1 cells were affixed to the wall after 24 h of PMA treatment and subsequently co-cultured with tumoral CM (Figure 4D). We then constructed a stable cell line overexpressing C2 in A375 (Figure 4E,F and Appendix A). In educated C2-CM macrophages, the RNA expressions of M1-associated CD86, IL12B, and IL23A were significantly higher. Although there was no difference in CD206, M2-associated IL10 levels were lower (Figure 4G). The overexpression of C2 in A375 cells increases the number of CD86 macrophages differentiated from M0 status cells (Figure 4H,I). In addition, we investigated the immunocytochemistry of the crawls, and the CD86+CD163− cells showed a significantly higher cell count compared to the control (Figure 4J,K). These results showed that C2-positive melanoma cells could promote M1 macrophage polarization.

### 3.5. C2 Influences the Formation of Tertiary Lymphoid Structures

C2 is associated with other beneficial immune cells besides macrophages. Tertiary lymphoid structures (TLSs) that form in non-lymph node tissues and whose formation in tumor tissues indicates a favorable prognosis and a strong response to immunotherapy. Considering the positive correlations observed between activated and effector memory CD8+ T cells, activated CD4+ T cells, and B cells and C2 expression levels (Figure 5A), we hypothesize a potential association between C2 and TLSs. The tumor infiltrations estimated by TIMER2 were all in general agreement with the aforementioned results (Figure 5B). Known molecular indicators of TLSs’ emergence include increased expression of CXCL12, CXCL13, CCL19, CXCR5, and LAMP3 [24]. We noticed that each of the above exhibited a positive connection with C2 levels in the transcriptome analysis (Figure 5C and Appendix A). Consecutive sections from 21 individuals were examined, and the overall TLS positivity rate was 38.1%. Only one patient in the C2-negative group developed TLSs, but C2 expression was strongly linked with the emergence of TLSs in 63.6% of the cases (Figure 5D,E). It is highly likely that C2 makes a positive contribution to the formation of TLSs in the TME.

## 4. Discussion

In this study, we compared tumor samples before and after nivolumab treatment and found that complement C2 levels continuously increase after effective treatment and patients with high levels of C2 after drug administration have longer overall survival. A high baseline before treatment also predicted good survival outcomes. We believe that the close relationship between C2 and the tumor immune environment is the reason for this effect. Tumoral C2 promotes macrophage polarization towards the M1 phenotype in vivo and in vitro. Additionally, it also contributes to the development of TLSs.

Component C2 is an inflammation protein that functions as part of the complement classical pathway [25]. C2 is split into C2a and C2b by activated C1. The C3 or C5 convertase is then produced by the serine proteinase C2a combined with complement factor 4b [26]. Some autoimmune diseases have been linked to C2 deficiency [27], and SNPs in C2 have been connected to altered susceptibility to age-related macular degeneration [28]. In the literature on cancer and complements, studies on C2 are almost absent. The sole study that did not elaborate in depth on C2 as a positive predictive factor in hepatocellular carcinoma was consistent with the results of the present pan-cancer analysis [29]. Moreover, ulceration, a predictive tissue biomarker for melanoma in tissue [30], has been incorporated into the current American Joint Committee on Cancer system [31]. Ulceration indicates a poorer prognosis and higher tumor stage. The expression level of C2 is negatively correlated with ulcer formation, suggesting that patients with high C2 expression have better prognostic outcomes. In this study, we utilized our center’s data in conjunction with multiple public databases and obtained consistent survival results in melanoma.

Melanoma is one of the most hazardous malignancies, and its incidence is on the rise worldwide [32]. Melanoma is the most immunogenic tumor, and it is inextricably linked to tumor immunity. In [33], mast cells were purified from melanoma skin biopsies for sequencing, and higher levels of both the mast cell-specific marker TPSAB1 and C3 expression, which share a significant association, predicted worse melanoma survival outcomes. This result demonstrates that the complement’s role in melanoma is mediated through immune cells in the TME. Our study revealed that although melanoma cell lines overexpressing C2 did not affect proliferation compared to the control, tumor growth was significantly restricted in the C2 overexpression group in a mouse model, characterized by an increased ratio of M1 to M2 macrophages. Activated M1 macrophages undergo phagocytosis and deliver antigens to T cells to trigger an adaptive immune response while also eliminating tumor cells [34]. CD8+ T cells and activated CD4+ T cells had significantly higher levels of infiltration in the high-C2 level samples in this study.

We compiled information from previously reported research to provide a complete picture of the prognostic impact of the principal immune cell subsets [3]. Previous studies have shown that the abundance of lymphocytes in melanoma is frequently a reliable predictive factor. Better patient prognoses are linked to T cells (including CD3, CD8, and CD4) and B cells (CD20), although Treg is mainly unrelated to prognosis [35,36,37,38,39]. B cells make up the majority of the adaptive immune system, along with T cells [40]. TLSs, which have been seen in several forms of cancer, including melanoma [41,42], are where B cells are located. TLSs might therefore improve antigen presentation, boost cytokine-mediated signaling, release tumor-specific antibodies, and improve prognosis [43]. We enriched the same pathways in the functional enrichment analysis of bulk RNA sequencing. The immune cell infiltration statuses and tissue architectures of patients in the high C2 expression group were outstanding, allowing for prolonged survival for this population. C2 can trigger the immune response in the TME. Macrophage phagocytosis and activation of tumor-infiltrating lymphocytes provide resistance against tumor cells through enhancing cytotoxicity and generating cytokines. The immune response may inhibit tumor development and spread, improving prognosis. Furthermore, we do not exclude the possibility that C2′s protective nature is connected to the responsiveness of therapies. C2 expression levels exhibited a negative relationship with the IC50 of anticancer medicines such as Trametinib, Temozolomide, Dacarbazine, and carboplatin, as determined using OncoPredict (Appendix A). These drugs may have a stronger inhibitory effect on the encoded protein, slowing tumor progression and improving patient prognosis. The TIDE algorithm, which predicts the efficacy of immunotherapy in patients, was also employed in this study. The lower the TIDE score, the greater the likelihood of benefit for the patient. We divided patients according to C2 expression level and found that the samples with high C2 expression had lower TIDE scores, indicating a greater likelihood of benefiting from immunotherapy (Appendix A).

During our bioinformatics analysis, we found that C2 is not only expressed in tumor cells but also present in macrophages. In the physiological state, the complement family components are produced mainly by hepatocytes and macrophages [44]. Macrophages not only generate C2 but also other members of the complement family. We speculate that macrophage-derived C2 may also have some effect on the tumor microenvironment, and we will further investigate this relationship in future studies.

This study was based on bioinformatics, and its results were validated by basic experiments. However, the direct mechanism between C2 and the TME remains to be confirmed in future work. In addition, melanoma is an important disease for basic and translational research aiming to improve cancer prognosis. Cancer research has made significant advances in the last decade with the introduction of treatments such as chemotherapy, targeted therapies, and immunotherapy [45]. In the future, we will explore the combination of immunotherapy with other drugs, including interferon, to enhance the immunotherapy’s efficacy. However, treatment response varies due to individual differences between patients, and the treatment of melanoma remains a challenge. C2 itself is an immune factor and may thus have the potential to be used in combination with immunotherapy to achieve greater efficacy and benefit more patients.

## 5. Conclusions

After effective immunotherapy, the upregulation of C2 expression, which is closely related to the tumor microenvironment, predicts better therapeutic efficacy and prognosis. C2 is a tumor suppressor gene in melanoma. It promotes the conversion of macrophages to the M1 subtype both in vivo and in vitro. It also has a positive correlation with the formation of TLSs. In clinical practice, complement C2 can predict a better immune status, which can serve as a prognostic indicator and dynamically monitor the efficacy of immunotherapy.

## Figures and Tables

**Figure 1 cancers-16-00908-f001:**
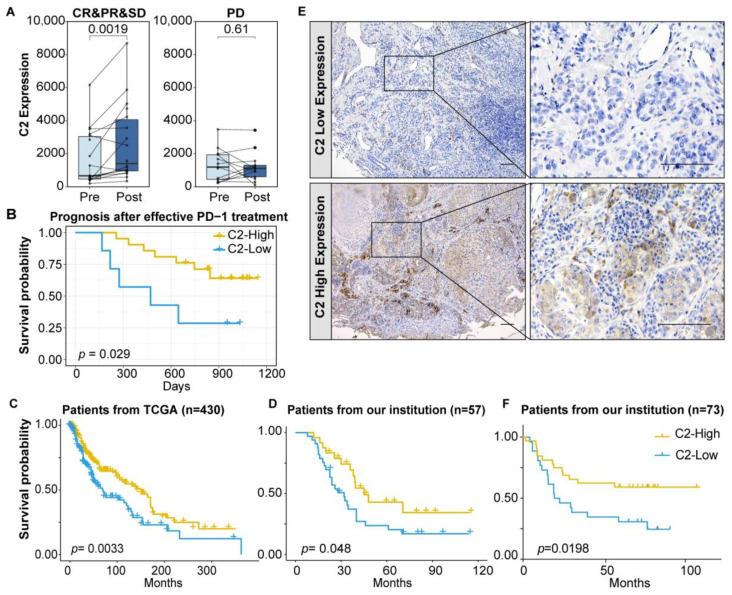
The clinical value of Complement C2 in melanoma. (**A**) Differential analysis of paired samples between pre- and post-nivolumab treatment in the CA209-038 study. (**B**) Kaplan–Meier analysis focused on C2 levels post nivolumab treatment. (**C**) Kaplan–Meier OS for different levels of C2 is shown based on the log-rank statistic in TCGA. (**D**) Kaplan–Meier OS for different levels of C2 in the inner institution. (**E**) Expression of C2 at high and low levels in melanoma tissues. Scale bar, 100 μm. (**F**) Prognostic significance of C2 expression in a series of 73 cases of melanoma based on IHC.

**Figure 2 cancers-16-00908-f002:**
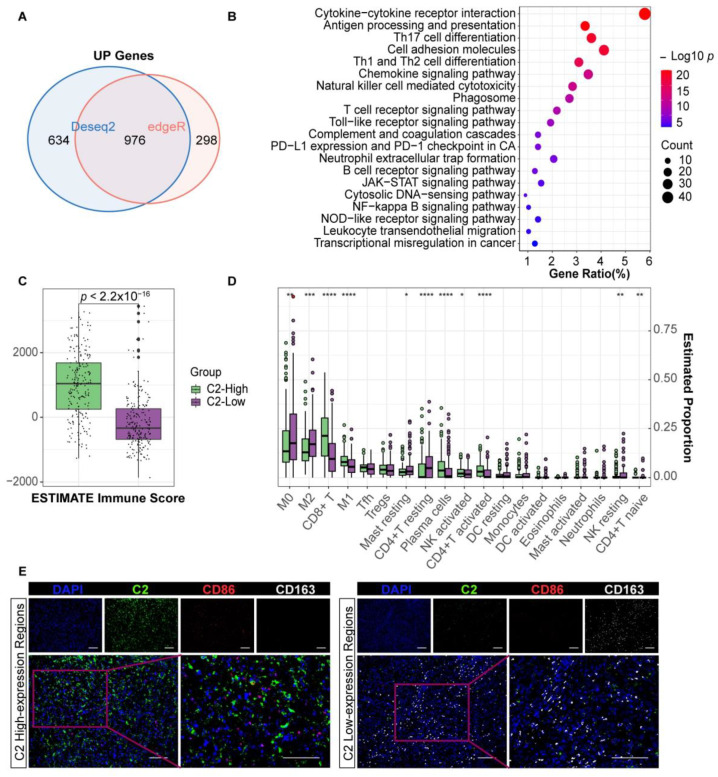
Complement C2 implies a more advantageous immune status. (**A**) Venn diagram presenting upregulated genes in Deseq2 and edgeR methods. (**B**) The most significant KEGG pathways are classified according to their *p*-value for upregulated gene sets with C2. (**C**) Immune score in different C2 expression groups. (**D**) Analysis of immune cell subtypes in melanoma by CIBERSORT. * *p* < 0.05, ** *p* < 0.01, *** *p* < 0.001, **** *p* < 0.0001. (**E**) Immunofluorescence staining (**left**) shows the aggregation of CD86+ macrophages (M1-like) in the C2-enriched region and almost no CD163+ macrophages. The (**right**) image shows CD163+ macrophages (M2-like) scattering in the C2-deficient region. Scale bar, 100 μm.

**Figure 3 cancers-16-00908-f003:**
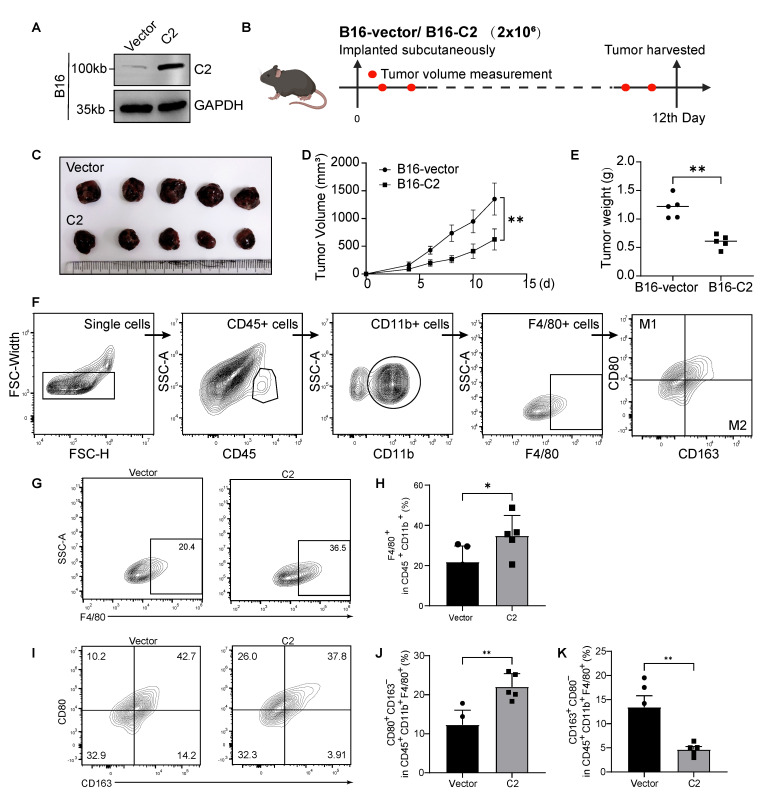
C2 inhibits melanoma growth and regulates the M1/M2 macrophage ratio in vivo. (**A**) Western blot of stable C2 overexpressing in B16 murine cell line. The uncropped bolts are shown in Appendix A. (**B**) Immunocompetent C57BL/6 mice were subcutaneously inoculated with B16-vector or B16-C2. Mice were sacrificed on the twelfth day after the injection. (**C**) Tumors at the endpoint of the experiment are shown. (**D**) Tumor growth curves of the two groups. Data are presented as mean ± SD between groups. (**E**) (*n* = 5 mice per group), tumor weight. (**F**) Flow cytometric analysis of macrophages (CD45+CD11b+F4/80+), M1 macrophages (CD45+CD11b+F4/80+CD80+CD206−), M2 macrophages (CD45+CD11b+F4/80+CD80−CD206+) in tumors. (**G**,**H**) Flow cytometric analysis and quantification of F4/80+ macrophages within CD45+CD11b+ fraction in C2 and vector tumors. (**I**–**K**) Flow cytometric analysis and quantification of M1 and M2 macrophages in C2 and vector tumors. *, *p* < 0.05; **, *p* < 0.01.

**Figure 4 cancers-16-00908-f004:**
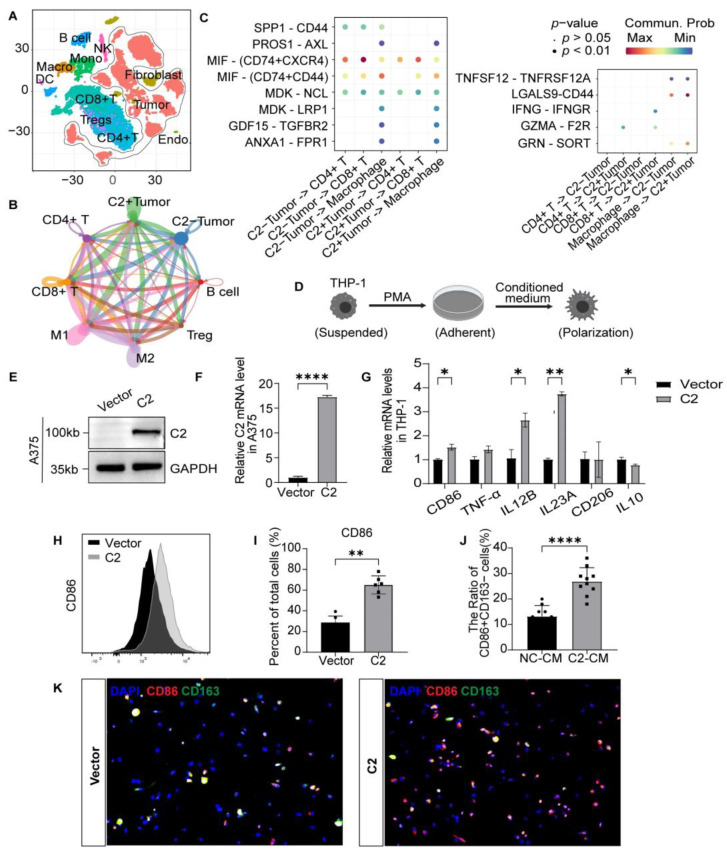
Tumoral C2 promotes macrophage differentiation into the M1 subtype in vitro. (**A**) t-SNE projection of single cells derived from melanoma patients colored by cell clusters. (**B**) Cellular communication between tumor and immune cells. (**C**) Bubble plot showing ligand–receptor interactions between C2+/C2− tumor cells and immune cells. Columns represent each cell–cell interaction pair, and rows define ligand–receptor pairs. (**D**) Tumoral conditioned medium co-cultured with the THP-1 cell line. (**E**) Western blot of stable C2 overexpressing in A375 human cell line. The uncropped bolts are shown in Appendix A. (**F**) Validation of RT-qPCR of overexpressed stable lines. (**G**) Relative gene expression of M1 marker (CD86, TNF, IL12B, IL23A) and M2 marker (CD206, IL10). (**H**,**I**) Flow cytometric analysis and quantification of CD86+ macrophages in C2-CM and vector-CM treatment. (**J**,**K**) Quantification of CD86+CD163− macrophages based on immunocytochemistry. *, *p* < 0.05 **, *p* < 0.01 ****, *p* < 0.0001.

**Figure 5 cancers-16-00908-f005:**
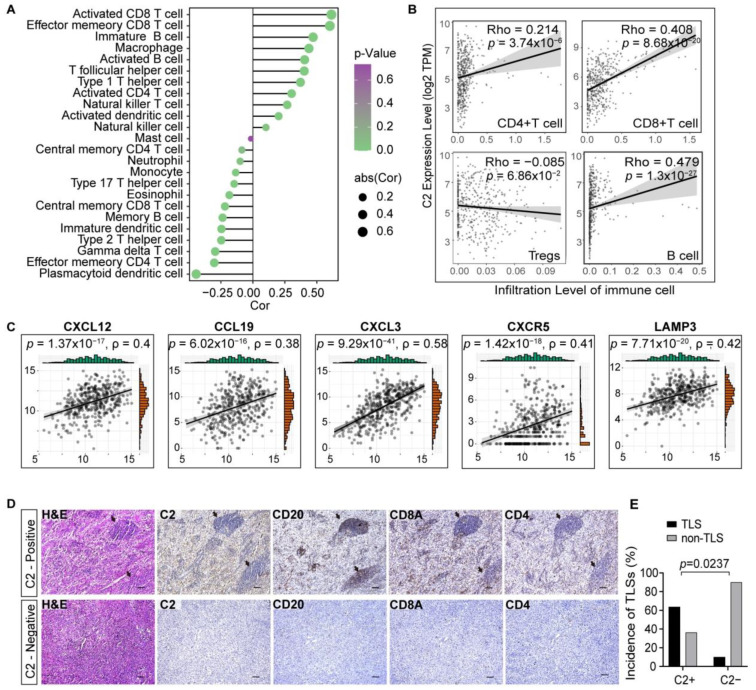
Tertiary lymphoid structures in the C2-enriched region. (**A**) Lollipop plot depicting Spearman’s correlation coefficients between C2 expression and immune cell types calculated by ssGSEA in the TCGA cohort. (**B**) Correlation between C2 and certain immune cells estimated by TIMER2. (**C**) Correlation analysis between C2 and the standard markers of TLSs. (**D**,**E**) Continuous histopathological sections illustrating TLS structures in different regions and statistical analysis. Scale bar, 100 μm.

**Table 1 cancers-16-00908-t001:** Characteristics of melanoma patients between low and high C2 expression groups in TCGA dataset.

Variables	TCGA Cohort (*n* = 326)	High C2 Expression Group	*p*-Value
Age (years)			0.848
<55	114 (35.0%)	58 (35.6%)	
≥55	211 (64.7%)	105 (64.4%)	
NA	1 (0.3%)		
Gender			0.172
Male	200 (61.3%)	94 (57.7%)	
Female	126 (38.7%)	69 (42.3%)	
Stage			0.566
I–II	197 (60.4%)	101 (62.0%)	
III–IV	123 (37.7%)	59 (36.2%)	
NA	6 (1.9%)	3 (1.8%)	
Breslow (mm)			0.065
<2	110 (33.8%)	63 (38.7%)	
≥2	209 (64.1%)	97 (59.5%)	
NA	7 (2.1%)	3 (1.8%)	
Clark level			0.106
I–III	88 (27.0%)	52 (31.9%)	
IV–V	199 (61.0%)	97 (59.5%)	
NA	39 (12.0%)	14 (8.6%)	
Ulceration			0.045 *
No	129 (39.6%)	71 (43.5%)	
Yes	151 (46.3%)	65 (39.9%)	
NA	46 (14.1%)	27 (16.6%)	
Mitotic index (/mm^2^)			0.625
≤6	101 (31.0%)	50 (30.7%)	
>6	56 (17.2%)	30 (18.4%)	
NA	169 (51.8%)	83 (50.9%)	
C2 expression			
High	163 (50%)		
Low	163 (50%)		

* *p*-value < 0.05.

**Table 2 cancers-16-00908-t002:** Univariate and multivariate analysis of C2 expression for overall survival.

Characteristics	Univariate Cox Regression			Multivariate Cox Regression		
	HR	95%CI	*p*	HR	95%CI	*p*
Age	1.94	1.29–2.92	0.001	2.20	1.13–4.28	0.020 *
(≥55 vs. <55)						
Breslow	1.71	1.15–2.54	0.008	0.97	0.47–1.98	0.926
(≥2 vs. <2)						
C2	0.63	0.43–0.93	0.019	0.53	0.28–0.99	0.047 *
(High vs. Low)						
Clark	1.98	1.29–3.05	0.002	1.12	0.57–2.21	0.748
(IV–V vs. I–III)						
Gender	0.97	0.64–1.45	0.868			
(Female vs. Male)						
Mitotic	1.67	1–2.8	0.051	1.7	0.84–3.43	0.138
(>6 vs. ≤6)						
Stage	2.44	1.61–3.69	0.000	2.93	1.58–5.44	0.001 *
(III–IV vs. I–II)						
Ulceration	2.13	1.38–3.27	0.001	1.34	0.72–2.5	0.351
(Yes vs. No)						

* *p*-value < 0.05.

**Table 3 cancers-16-00908-t003:** Correlation analysis of C2 protein expression levels with CD86 and CD163.

		C2 Expression			
		Low	High	Total	*p*
	−	7	2	9	
CD86	+	3	9	12	0.03
	Total	10	11	21	
	−/+	1	7	8	
CD163	++	9	4	13	0.0237
	Total	10	11	21	

“−” represents <10 cells; “+” represents 10–49 cells; “++” represents >50 cells.

## Data Availability

The raw bulk RNA-seq data in the internal institute have been deposited in the GEO database under code GSE215121. The GEO database contains the publicly accessible data utilized under the registration codes GSE65904 and GSE174401.

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
