# Peer review of "Tumoral C2 Regulates the Tumor Microenvironment by Increasing the Ratio of M1/M2 Macrophages and Tertiary Lymphoid Structures to Improve Prognosis in Melanoma"

_cancers, 2024, doi:10.3390/cancers16050908_

Round 1

Reviewer 1 Report

Comments and Suggestions for Authors

The manuscript by Gengpu Zhang et al. reports a previously unknown role for the C2 complement component in shaping the tumor microenvironment in cancer melanoma by acting on the M1/M2 ratio of tumor-infiltrating macrophage population. The manuscript is clearly written and the experimental design very sound. Their results are supported by a solid bioinformatic analysis, corroborated by strong experimental evidences. The statistical methods applied are correct. The key innovation of their work stems from the demonstration for the first time of a role for the C2 complement component in the repression of melanoma tumor growth in both patients-derived data and in a syngeneic in vivo model. Based on these evaluation, I think the manuscript is by no doubt acceptable for publication in Cancers.

Minor points:

- Although the results from the syngeneic mouse model are solid and clear-cut, end-point analysis of C2 expression levels in the tumors could further corroborate the observed correlation between C2 expression and tumor growth retardation

- In section 3.4, the authors state that "the communication between C2-positive tumor cell and immune cells is more prominent than that of C2-negative cells". This seems to suggest that tumor cells represent the main source of C2 protein in vivo. Could it be that other cells in the tumor microenvironment besides (or alternatively to) cancer cells might be the source of C2? This issue could be addressed in the discussion section.

Comments on the Quality of English Language

The are a few misspellings here and there.

Reviewer 2 Report

Comments and Suggestions for Authors

In this manuscript, Zhang and collaborators investigated the role of C2 complement in melanoma. Using subcutaneous tumorigenic models in C57BL/6 mice, they found that C2-overexpression tumor cells exhibited significantly smaller sizes correlated with enhanced recruitment of macrophages, particularly of the M1 subtype. Moreover, scRNA sequencing analysis revealed that C2-positive tumor cells exhibited enhanced connection with immune cells. Using C2-overexpressing tumor cell supernatant co-cultured with macrophages, they also observed C2-mediated induction of M1 subtype polarization.

In general, although the research subject is of interest, the quality of images in Results section is very low and in some cases, it is quite difficult to read and understand the data. All figures should be uploaded with high quality.

Specific comments:

Fig 1: Very low-quality images. It is almost impossible to read the data. In addition, at line 217 “ulceration formation which is a carcinogenic process”, a reference regarding the ulceration as cancerogenic factor in melanoma should be added.

Fig 2: CD163 is not a specific marker for M1 macrophages as demonstrated by Molgora M et al (PMID: 32783918). Authors should validate their conclusion with CD206 and TREM2 co-staining.

Fig 3: flow cytometry analysis using CD206 should be added.

Fig 4: Line 312 “Human THP-1 cells (M0 status)”: what means? Authors should demonstrate that THP1 are in M0 status. In addition, human monocytic leukemia cell line THP1 is not the best model to study the role of C2 on macrophages polarization. I suggest validating these data using human PBMC-derived macrophages.

Fig 5: Data presented in this figure are not convincing. According to fig 5D, is very difficult to distinguish follicles structure inside the lymph node. In addition, authors should remove “figure 4” on the top.

Comments on the Quality of English Language

English editing require an extensive revision

Round 2

Reviewer 2 Report

Comments and Suggestions for Authors

Thank you to the authors to address the raised questions. The new version of the manuscript is improved.